# Retirements of professional tennis players in second- and third-tier tournaments on the ATP and WTA tours

**Maria Palau**[1], **Ernest Baiget** [2], **Jordi Cortés**[3], **Joan Martínez**[4], **Miguel Crespo** [5], **Martí Casals** [2,6,7] *

1 Universitat Oberta de Catalunya, Barcelona, Spain, 2 National Institute of Physical Education of Catalonia (INEFC), University of Barcelona, Barcelona, Spain, 3 Department of Statistics and Operations Research, Universitat Politècnica de Catalunya, Barcelona, Spain, 4 Girona Biomedical Research Institute-IDIBGI, Salt, Spain, 5 Development Department, International Tennis Federation, London, United Kingdom, 6 Sport and Physical Activity Studies Centre (CEEAF), Faculty of Medicine, University of Vic-Central University of Catalonia (UVic-UCC), Barcelona, Spain, 7 Sport Performance Analysis Research Group, University of Vic-Central University of Catalonia (UVic-UCC), Barcelona, Spain

* marticasals@gmail.com

**Data Availability Statement:** https://github.com/mariapalau/tennis-retirements.

**Funding:** This research was funded by the Ministerio de Ciencia e Innovación (Spain)

## Abstract

The demands of professional tennis, including physical and psychological aspects, contribute to the frequency of retirements at elite levels of the sport. The aim of this study was to analyze epidemiological patterns and risk factors associated with retirements in previous ATP and WTA Tour tournaments. A retrospective cohort study was conducted. This study focused on previous ATP and WTA Tour tournaments. The ATP database encompassed 584,806 matches, while the WTA database included 267,380 matches. To assess retirements, potential risk factors such as playing surface, tournament category, match round, and player age were analyzed. Incidence rates were calculated for the period between 1978–2019 for men and 1994–2018 for women. The overall incidence rate was 1.56 (95% CI: 1.54, 1.59) and 1.36 (95%CI: 1.33, 1.39) retirements per 1000 games played in male and female competitions, respectively. Retirements increased over the years. Higher incidence rates were observed on hard (1.59 [95%CI: 1.56, 1.63] and 1.39 [95%CI: 1.34, 1.44]) and clay (1.60 [95%CI: 1.57, 1.63] and 1.36 [95%CI: 1.32, 1.41]) compared to grass courts (0.79 [95%CI: 0.65, 0.94] and 1.06 [95%CI: 0.88, 1.27]). Risk factors differed by gender, with tournament category significant in males (IRR: 1.23 [95%CI: 1.19, 1.28] in ITF vs ATP) and match round in females (IRR: 0.92 [95%CI: 0.88, 0.98] in preliminary vs final). This study provides valuable insights for coaches, players, support teams, and epidemiologists regarding retirements and associated risk factors in previous ATP and WTA Tour tournaments, contributing to injury prevention strategies.

## Introduction

Tennis is one of the most popular sports worldwide, with over 75 million participants from all ages and levels, and a discipline that attracts millions of fans to the most important

(PID2019- 104830RB-I00) and the Departament de Recerca i Universitats de la Generalitat de Catalunya (Spain) [2021 SGR 01421 (GRBIO)]. The funders had no role in study design, data collection and analysis, decision to publish, or preparation of the manuscript.

tournaments every year [1–3]. In 2018, the US tennis economy was valued at $6.19 billion [4], with the popularity of the sport witnessing a significant increase in overall participation globally during the COVID-19 pandemic [5].

The popularity of tennis, the particularities of the sport itself, and the evolution of the game is a topic of interest in the field of sports medicine and health in general, especially in recent years [6, 7]. In the last decades, competitive tennis has experienced a complete transformation in many facets that define the game such as technology, administration, and sport science, among others. As for the professional side of the game, the current physical, technical, tactical, and mental demands are highly challenging for the top players. In this context, sport science advancements applied to training and competition have also impacted the way in which tennis is taught and played [8, 9]. Tennis has evolved from a technical, tactical discipline based on style and finesse and played with wooden rackets, to the current fast paced, explosive sport based on power, strength, and speed, and played with lighter, stronger and bigger rackets [10]. During the last decades there has been a tendency of an increased ball speed and taller players with professional players able to generate higher amounts of power behind their shots (11,12). From a physiological and physical perspective, these changes and the intermittent nature of the game itself require professional tennis players to possess a combination of anaerobic skills together with high aerobic capabilities, and to be proficient in most components of physical fitness, as well as in tactical, technical and psychological areas [11, 12]. Moreover, the physiology of tennis is considerably complex because of the start-stop intermittent nature of the game (which alternates short bursts of high-intensity exercise and short recovery bouts) and the uncertainty of the duration of matches [12, 13]. These particularities, as well as other factors such as the changing playing surface, the different ball types, the variable environmental conditions, or the various playing styles of the opponents, which are known to affect the physical and psychological requirements of tennis match play, lead to significant physiological stress, which can affect the performance of the player [14–16]. Competition male and female tennis players have to face powerful strokes during matches up to 3 hours, with mean rallies durations and covered distances of 5.5 and 6.4 s and 9.6 m and 8.2 m [9]. Each rally requires to perform an average of 4–5 short, repetitive, and multidirectional high-intensity changes of direction, covering approximately 3 m, and resulting in a total of 250 to 400 changes of direction during the whole match [17]. Therefore, competition tennis players have to prepared to repeatedly over an extended period of time execute high-intensity actions (i.e., shots and displacements) and to recover fast from it with both aerobic and anaerobic metabolic demands [12].

These high physical requirements imposed on the professional players and the training loads they have to deal with to cope with this scenario, have produced an increase in the number and frequency of retirements and medical withdrawals of these players from competitions and the rise of a unique profile of injuries [18–20]. A meta-analysis of published articles from 1966 to 2005 reported an injury incidence that varied from 0.04 injuries/1000 hours to 3.0 injuries/1000 hours played [21]. Injuries and illness surveillance and epidemiological studies are fundamental elements of concerted efforts to protect the health of athletes and should be the first step in any injury prevention program [22, 23]. In recent decades, sports physicians have emphasized the importance of epidemiological research in sports medicine, as sport participation is associated with a certain injury risk. Knowing the incidence, patterns, and severity of injuries is a fundamental aspect in the development of adequate prevention strategies [24].

For over two decades, many epidemiological surveillance protocols have been carried out in multiple team sports and have been implemented regularly in national and international competitions [25–29]. Individual sports like tennis have also implemented epidemiological surveillance programs aimed to study the injury profile related to the game [21, 30]. To unify criteria and improve the quality of these surveillance studies, and following consensus

statements developed for many team sports [31–34], a consensus statement for tennis was published in 2009 including consistent definitions, data collection procedures and methods of reporting results in injury surveillance studies [35]. A new guideline was published in 2020 [36] as a tennis-specific extension of the 2020 IOC consensus statement [37] which updated the 2009 consensus statement on tennis medical conditions. This new consensus provides tennis-specific additions to the IOC consensus, highlighting the importance of monitoring the health of tennis players to contribute to their welfare as well as to reduce injury and illness. It details the baseline information necessary to collect, the injury and illness reporting and classification, and the match and training exposure, as well as the risk expressed as the number of injuries per 1000 hours and per 1000 games played [36].

Withdrawals, walkovers, and retirements in tennis can occur due to multiple reasons and in many circumstances. Withdrawals and walkovers happen when a player abandons competition before the tournament or the match, while retirements occur once the match has already begun. A recent article explored the withdrawals in professional tennis, considering all walkovers and retirements [38]. Commonly, retirements during professional tennis matches are closely related to injuries, heat stress, illnesses, and hydration [39–42]. Other factors that may affect the incidence and type of injuries and the retirements in professional matches are the playing surface [43], age and gender [44], previous injury [45], the tournament type and round, the players' ranking [46] and the volume of play [47]. A considerable number of tennis epidemiology studies on retirement and withdrawals of players focus on the ATP and the WTA Tours tournaments [20, 39, 46, 48–51]. There have also been many studies on elite junior tennis players [52–55]. However, epidemiological research focusing only on professional competitions below the ATP and the WTA Tours, such as the ATP Challenger Tour and ITF Men's World Tennis Tour for men, and WTA 125 Tournaments and ITF Women's World Tennis Tour for women, is limited [20, 56, 57].

The first ATP Challenger events were held in 1978, and the ITF Circuit started in 1990, consisting of Satellite tournaments, until 1998 when the ITF introduced Futures tournaments [58, 59]. The first season of the ITF Women's Circuit took place in 1994, and it was the level immediately below the WTA Tour, until 2012 when the WTA introduced the WTA 125 tournaments as an intermediate level [60, 61]. These professional tournaments are crucial in the careers of any promising tennis player since they provide a very important competitive experience to players aspiring to compete in the ATP and WTA Tours.

This study focuses on the analysis of the professional tennis tournaments that are not part of the ATP or WTA Tour. The tournaments analyzed were part of the ATP Challenger Tour and ITF Men's World Tennis Tour for men, and WTA 125 Tournaments and ITF Women's World Tennis Tour for women. For the purposes of the study, it was considered that retirements implied that the match was being played until the withdrawal occurred as opposed to a walkover, which can occur due to other reasons apart from medical issues. Therefore, the aim of this study is to describe the epidemiological retirement patterns of professional tennis players from international tournaments behind ATP and WTA Tour over the years. This study also seeks to analyze the associated factors and characteristics in tennis match play that influence the retirements in professional players.

## Material and methods

### Procedures

A retrospective cohort study was conducted to analyze the epidemiological pattern of retirements for both male and female professional tennis players in tournaments of the ATP

Challenger Tour and ITF Men's World Tennis Tour for men, and WTA 125 Tournaments and ITF Women's World Tennis Tour for women.

## Ethics statement

The study involving ATP and WTA matches, being a retrospective cohort analysis based solely on publicly available data, did not involve direct interaction with human subjects or animals, and therefore, ethical approval was not applicable for this research.

## Subjects

The dataset of tennis matches used in this study is available at GitHub (https://github.com/skoval/deuce/tree/master/data) or at the R package "Deuce", which provided easy access to a rich set of online data on professional tennis from web scraping of official ATP, WTA and ITF data [62]. The databases used were "atp_matches.RData" and "wta_matches.RData", with both datasets containing 72 variables each one. To analyze the matches from previous ATP Tour tournaments, the tournaments "Challenger" and "Futures" were selected. Following the same criteria in the WTA, the tournaments "Challenger", "Satellite", "C10", "C20", "C25," "C50", "C75", "C100", and "C125" were selected. The selection of the tournaments and their correspondence with the data in the database is shown in Table 1. Having selected the tournaments of interest, the ATP database contains 584,806 matches, and the WTA database contains 267,380 matches.

## Research instrument

As previously mentioned, ATP and WTA databases have seventy-two variables. The reasons behind incomplete or non-played tennis matches are shown in Table 2 [62, 63]. This study only focuses on retirements, that is, on matches that started but did not finish for any reason.

**Table 1. Classification of tournaments from professional tennis.** In gray, the tournaments selected for the study.

| | Tour | | Tournament | Deuce: Tournament_level |
|---|---|---|---|---|
| **ATP** | ATP Tour | Top-tier Tour | Grand Slam | Grand Slams |
| | | | ATP Finals | Tour Finals |
| | | | ATP Masters 1000 | Masters |
| | | | ATP 500 | 250 or 500 |
| | | | ATP 250 | 250 or 500 |
| | ATP Challenger Tour | Second-tier Tour | Challenger | Challenger |
| | ITF Men's World Tennis Tour | Third-tier Tour | Futures | Futures |
| | Team Events | - - - - | Davis Cup | Davis Cup |
| | | | Olympics | - - - - |
| **WTA** | WTA Tour | Top-tier Tour | Grand Slam | Grand Slams |
| | | | WTA Finals | Historical |
| | | | WTA 1000 | Premier Mandatory + Premier Five + Tier I + Tier II |
| | | | WTA 500 | Premier |
| | | | WTA 250 | International + Tier III + Tier IV + Tier V |
| | | | Past tournaments of WTA Tour | Historical |
| | WTA 125 Tournaments | Second-tier Tour | WTA 125 | Challenger + C125 |
| | ITF Women's World Tennis Tour | Third-tier Tour | ITF Women's Circuit | Satellite + C10 + C20 + C25 + C50 + C75 + C100 |
| | Team Events | - - - - | Davis Cup | Davis Cup |
| | | | Olympics | Olympics |

**Table 2. Types of abandonments in tennis.**

| Term | Abbreviation | Explanation |
|------|--------------|-------------|
| WITHDRAWAL | WD | A player withdraws before the tournament starts. |
| WALKOVER | W/O | A player withdraws before the match starts. |
| RETIREMENT | RET | A player withdraws during the match. |
| DEFAULT | DEF | A player is disqualified due to a violation of the code of conduct. |
| TIME | TIME | A match is suspended temporarily due to external factors, such as weather or technical problems. |

The outcome variable is "Retirement" with two possible options: Yes or No.

The covariables defining the match and tournament characteristics are shown in Table 3.

The variables defining the players' features are shown in Table 4.

The R workspace which contains the data frames that were used for the analysis of the data is available at https://github.com/mariapalau/tennis-retirements.

## Statistical analysis

**Data wrangling and exploratory data analysis.** Before the exploratory data analysis, a process of data wrangling was conducted to generate new variables and the update of the dataset's scores. On one hand, new variables were created using the preexistent ones as a reference for the new calculations. On the other hand, the data wrangling process involved an update of

**Table 3. Description and type of the variables of interest, regarding the match and tournament.**

| Variable | Description |
|----------|-------------|
| Tournament Category | A new variable, classifying the tournaments in four categories:<br>• ATP Challenger Tour<br>• ITF Men's World Tennis Tour<br>• WTA 125 Tournaments<br>• ITF Women's World Tennis Tour |
| Tournament Level | A character description of the tournament level. The tournaments included are specified in the Table 1. |
| Year | Numeric value of the year the match occurred. |
| Surface | A character description of the court surface (Carpet, clay, grass, or hard). |
| Best Of | A numeric value indicating the match format. The total number of sets in ATP can be 3 or 5, whereas in WTA it is always 3. |
| Round Level | A new variable, classifying the match round into three levels:<br>• Qualifying Round. In most of the tournaments, players need to qualify for the main draw and play in a qualifying draw through different qualifying rounds. This includes Q1, Q2, Q3, and Q4.<br>• Preliminary Round. This includes R64, R32, and R16 of the main draw, where there are 64, 32, and 16 players remaining in competition, respectively. In some tournaments (ATP and WTA Finals (Masters)), players are organized in groups, and they play against each other in a round robin format (RR), where the ones with the most wins progress to the next round.<br>• Final Round. The final phases of the tournaments' main draws include quarter-finals (QF), semi-finals (SF), and finals (F). |
| Score | The result of the match. |
| Games | A new variable, with the total number of games of the match, summing the variables W1, W2, W3, W4, W5, L1, L2, L3, L4, and L5. The variables W1-W5 indicate the number of games won by the winner. The variables L1-L5 indicate the number of games won by the loser. |
| Match Outcome | A new variable, indicating whether the match has been completed or not, and if not the reason why (RET, W/O or DEF). From this variable, the outcome variable "Retirement" has been created (RET). |

**Table 4. Description and type of the variables of interest, regarding the players.**

| Variable | Description |
| --- | --- |
| Hand dominance | A character value indicating the handedness of the winner and loser (R[1], L[2], A[3], or U[4]). |
| Winner age | A numeric of the winner's age at the time of the match. |
| Loser age | A numeric of the loser's age at the time of the match. |
| Difference age | A new variable, indicating the difference between the winner's age and the loser's age. |
| Sum age | A new variable, indicating the sum of the winner's age and the loser's age. |
| Mean age | A new variable, indicating the mean between the winner's age and the loser's age. |
| Winner rank | A numeric of the winner's rank at the time of the match. |
| Loser rank | A numeric of the loser's rank at the time of the match. |
| Difference Rank | A new variable, indicating the difference between the winner's rank and the loser's rank. |
| Sex | A new variable, indicating the sex of the players, being male in ATP and female in WTA. |

[1] Right,

[2] Left,

[3] Ambidextrous,

[4] Unknown

the database. This step was based on the exploration of missing and miscellaneous match results and the retrieval, when possible, of data from official sources of ATP, WTA and ITF sites on Internet (https://www.atptour.com/en/, https://www.wtatennis.com/, https://www.itftennis.com/en/). After examining the variable "Score", those registries which did not have a match result, which had a value that did not fit the categories of a complete match, default, walkover or retirement, or which had a result that was not in the proper format, were selected. The results of these matches were searched in the results archive of ATP, WTA and ITF on Internet, searching by year, tournament, round and players. The scores found were retrieved and modified in the database, obtaining a more complete dataset and avoiding a large number of missing or wrong values.

Once the dataset's scores were updated, the variables "Games" and "Retirement" were also updated, so that the number of games and the number of retirements were consistent with the new match results. The final versions of the ATP and WTA databases used for further analysis contain 12 variables, with the outcome variable *retirement* and other covariables of interest describing the match and players' characteristics. The covariables selected were the following: *tournament_category*, *year*, *surface*, *best_of*, *round_level*, *score*, *games*, *match_outcome*, *mean_age*, *dif_rank*, and *sex*. The focus of this study lies on retirements, and therefore, the outcome variable assembles the match outcomes in two groups, considering whether or not a retirement has occurred.

**Epidemiology measures.** In sports medicine, epidemiological research is of utmost importance for injury prevention (24). Therefore, surveillance programs with consistent methods for data collection and reporting measures are fundamental in every sport [35–37]. As the outcome variable in this study is the retirements occurred during matches, incidence measures are reported, since incidence-based measures usually represent more appropriate outcomes for sudden-onset conditions [37]. Therefore, the International Olympic Committee recommends registering exposure in time, as well as recording the number of matches, sets, games and/or points played, and expressing risk as the number of injuries per 1000 hours and per 1000 games played [36].

In this report, incidence rates are indicated. The outcome analyzed is the number of retirements, and the exposure is quantified as the number of games played as a measure of the time

during which the athletes are at risk of injury. Therefore, and following the guidelines of the International Olympic Committee consensus statement [37] and the tennis-specific extension [36], the incidence rates are expressed as the number of retirements per 1000 games played. In epidemiological studies, the report of measures of association and their precision is also key for appropriate evaluation of study results [64]. In sports epidemiology, relative measures are much more used than the absolute ones although STROBE statement for observational studies and the CONSORT statement for randomized trials recommend researchers to report both [65, 66].

Relative measures of association are estimated by presenting results as a ratio of two groups, whereas absolute measures are produced by subtracting risks or rates from two exposure groups. Therefore, relative measures aim to identify the strength of association between an exposure of interest and injury development, while absolute measures aim to identify how many more injuries are sustained in one group as compared with another group [64].

Following the STROBE and CONSORT statements, both relative and absolute measures are presented in this report. The incidence rate ratio (IRR) is a relative measure of association, calculated as the ratio of the incidence rate of one group divided by the rate of the reference group, and expressed in percentage. The risk difference (RD) is an absolute measure of association, calculated as the difference between the incidence rate of one group and the incidence rate of the reference group, and it is expressed as retirements per 1000 games played [64]. In this study, for each group of the covariables of interest, incidence rates, incidence rate ratios, and risk differences are reported, estimating their values with a 95% confidence interval. Incidence rates through the years for ATP and WTA databases are also presented in a form of a plot.

**Multivariable models.** At multivariable level, a generalized linear Poisson model was fitted. The model expression is:

$$\log(\lambda_i) = \log(g_i) + \alpha + \beta X_i + u_i$$

where $Y_i \sim Poisson(\lambda_i)$; $\lambda_i$ is the expected number of retirements for the i-th match; $g_i$ is the number of games of the i-th match, and $log(g_i)$ is the offset of this model; $X_i$ includes all independent variables of interest for the i-th match; $\alpha$ is the intercept of the model; $\beta$ represents the vector of coefficients associated to covariates; and $u_i$ is the error term.

Model selection was performed using a stepwise method based on the Akaike Information Criterion (AIC). ATP and WTA competitions are independent with relevant differences in the players' game styles. However, gender effect interaction was analyzed using an interaction plot, to statistically justify the creation of a different model for ATP and WTA databases. A couple of Poisson regression models were separately fitted for ATP and WTA tours. Linearity of the variable year was verified for both ATP and WTA databases. Both ATP and WTA Poisson final models were tested for overdispersion according to the method proposed by Kleiber & Zeileis [67]. In the ATP model, overdispersion was found. However, as the estimated dispersion parameter can be considered low (1.44) [68, 69] and the results of a negative binomial model did not greatly differ from the Poisson model, the latter is presented. In the WTA model, overdispersion did not occur (1.09), and again, the Poisson model is presented.

Measures of association were calculated using IRR with 95% CI. The significance level was set at $\alpha = 0.05$.

All analyses were performed using version 4.1.3 of the R statistical software [70]. The package *compareGroups* [71] was used to describe characteristics according to the presence of retirement. To calculate the incidence rates, the *epi.2by2* function of the package *epiR* [72] was mainly used, setting the argument *method* as *cohort.time*. Most of the graphics were obtained

**Table 5. Frequency of the match outcome of ATP/WTA database, with the number of observations (N) and the percentage over the total number of matches.**

| Match Outcome | N (ATP) | Percentage (%) (ATP) | N (WTA) | Percentage (%) (WTA) |
|---|---|---|---|---|
| Complete | 5629,80 | 96.27 | 2588,23 | 96.80 |
| Retirement | 1931,4 | 3.30 | 7306 | 2.73 |
| Walkover | 2263 | 0.39 | 1218 | 0.46 |
| Default | 235 | 0.04 | 33 | 0.01 |
| Unknown | 14 | 0.00 | 0 | 0.00 |

using *ggplot2* [73] package. The overdispersion were tested with the dispersion.test function in the package *AER* [67]. We used the anova function from package *stats* [70] to make comparisons between models.

The R data and code used for all the statistical analyses is available at https://github.com/mariapalau/tennis-retirements.

## Results

The summary of the match outcome in the ATP and WTA databases is shown in Table 5.

The ATP database contains 584,806 matches, 562,980 (96.27%) of which are completed matches and 19,314 (3.30%) are matches where a retirement occurred. The remaining 2,512 (0.43%) registries correspond to walkovers, defaults, or missing values for which were not possible to retrieve the result. As per the WTA database, it contains 267,380 matches, 258,823 (96.80%) of which are completed matches and 7,306 (2.73%) are matches where a retirement occurred. The remaining 1,251 (0.47%) registries correspond to walkovers or defaults.

Regarding the tournament category distribution in the ATP database, 73.35% of the cases are from the ITF Men's World Tennis Tour, and 26.65% from the ATP Challenger Tour matches. As per the surface of play, clay and hard courts are 52.16 and 42.00% of the cases, respectively, while carpet and grass surfaces being 4.68 and 1.16% of the matches, respectively. In terms of the round of the match, 98.03% of the observations are from the main draw rounds of the tournament, being 76.05% from the preliminary rounds and 21.98% from the final rounds. Qualifying draw matches are 1.96%.

Considering the tournament category in the WTA database, 99.81% of the observations are from the ITF Women's Tour, while only a 0.19% of the matches are from the WTA 125 Tournaments. As per the surface of play, clay or hard courts are 47.80 and 45.49% of the cases, respectively, while carpet and grass surfaces are 4.61 and 2.10%, respectively. Regarding the round of the match, 99.98% of the observations are from the main draw rounds of the tournament, the preliminary rounds being 77.31% and the final rounds 22.67%. Qualifying draw matches are 0.02%.

### Descriptive characteristics of ATP and WTA matches

A summary of the descriptives by retirement for the ATP and WTA databases is shown in Tables 6 and 7. The categorical variables are presented with the number of observations for each level and the percentage by row.

Regarding ATP Tour matches and considering the tournament category, the ATP Challenger Tour has a percentage of retirements of 2.94%, while the ITF Men's World Tennis Tour has a percentage of retirements of 3.44%. The surface of play variable presents a proportion of retirements that varies from 1.78% in grass courts to 3.38% in hard courts, and the percentage of retirements depending on the match round goes from 3.28 to 3.55%. The player's mean age

**Table 6. Summary descriptives table by retirement in ATP Tour matches.**

| Retirement | | |
| --- | --- | --- |
| | NO | YES |
| | *N = 565492* | *N = 19314* |
| **Tournament Category**: | | |
| ATP Challenger Tour | 1512,69 (97.06%) | 4577 (2.94%) |
| ITF Men's World Tennis Tour | 4142,23 (96.56%) | 1473,7 (3.44%) |
| **Surface**: | | |
| Carpet | 2666,4 (97.38%) | 717 (2.62%) |
| Clay | 2948,31 (96.66%) | 1017,6 (3.34%) |
| Grass | 6675 (98.22%) | 121 (1.78%) |
| Hard | 2373,22 (96.62%) | 8300 (3.38%) |
| **Round**: | | |
| Final Round | 1242,50 (96.65%) | 4307 (3.35%) |
| Preliminary Round | 4301,69 (96.72%) | 1460,0 (3.28%) |
| Qualifying Round | 1107,3 (96.45%) | 407 (3.55%) |
| **Mean of Ages** | 22.76 [21.02;24.76] * | 23.26 [21.42;25.26] * |

* Median [Q1; Q3]

variable, as normality cannot be assumed, is displayed with its median, and the first and third quartiles in square brackets. The median values are 22.76 years for the matches where no retirement occurred, and 23.26 years for the cases where a retirement occurred (Table 6).

Regarding the tournament category, the ITF Women's World Tennis Tour has a percentage of retirements of 2.73%, while the WTA 125 Tournaments have a percentage of retirements of 2.94%. As for the surface of play variable, the proportion of retirements varies from 2.21% in grass courts to 2.78% in hard courts. The match round presents a percentage of retirements that goes from 2.65 to 4.35%. The female tennis players mean age variable, as normality cannot

**Table 7. Summary descriptives table by retirement in WTA Tour matches.**

| Retirement | | |
| --- | --- | --- |
| | NO | YES |
| | *N = 260074* | *N = 7306* |
| **Tournament Category**: | | |
| WTA 125 Tournaments | 496 (97.06%) | 15 (2.94%) |
| ITF Women's World Tennis Tour | 2595,78 (97.27%) | 7291 (2.73%) |
| **Surface**: | | |
| Carpet | 1203,1 (97.62%) | 293 (2.38%) |
| Clay | 1243,07 (97.25%) | 3509 (2.75%) |
| Grass | 5482 (97.79%) | 124 (2.21%) |
| Hard | 1182,54 (97.22%) | 3380 (2.78%) |
| **Round**: | | |
| Final Round | 5881,3 (97.00%) | 1816 (3.00%) |
| Preliminary Round | 2012,17 (97.35%) | 5488 (2.65%) |
| Qualifying Round | 44 (95.65%) | 2 (4.35%) |
| **Mean of Ages** | 20.67 [19.03;22.60] * | 21.00 [19.32;22.98] * |

* Median [Q1; Q3]

be assumed, is displayed with its median, and the first and third quartiles in square brackets. The median values are 20.67 years for the matches where no retirement occurred, and 21.00 years for the observations where a retirement occurred (Table 7).

## Epidemiology of matches of ATP and WTA

The results of the epidemiological study with measures of frequency and association for the ATP and WTA database are presented below (Tables 8 and 9).

The evolution of the incidence rate for the ATP and WTA databases over the years of study (1978–2019 for ATP, and 1994–2018 for WTA) is shown in Figs 1 and 2, respectively. The incidence rates are expressed as number of retirements per 1000 games played.

In both cases, the evolution of the retirement incidence rate for the ATP and WTA databases shows a similar increasing trend. However, the incidence rates in ATP are higher than in WTA. The different epidemiological measures of frequency and association for the different groups of interest for ATP and WTA databases are shown below. The covariates *Tournament Category*, *Surface*, and *Round*, have been considered, and for each level incidence rates, incidence rate ratios, and risk differences are presented, with their 95% confidence interval. The

**Table 8. ATP incidence rates, incidence rate ratio (IRR), risk difference, and the 95% confidence intervals (CI95%), expressed per 1000 games played.**

| ATP | Retirements | Games | Incidence Rate (95%CI) | IRR (95%CI) | RD (95%CI) |
|---|---|---|---|---|---|
| **Tournament Category:** | | | | | |
| ATP Challenger Tour | 4577 | 3420,806 | 1.34 (1.30, 1.38) | 1 (Ref.) | 0 (Ref.) |
| ITF Men's World Tennis Tour | 14737 | 8922,704 | 1.65 (1.63, 1.68) | 1.23 (1.19, 1.28) | 0.31 (0.27, 0.36) |
| **Surface:** | | | | | |
| Grass | 121 | 1536,97 | 0.79 (0.65, 0.94) | 1 (Ref.) | 0 (Ref.) |
| Clay | 10176 | 6367,080 | 1.60 (1.57, 1.63) | 2.03 (1.70, 2.45) | 0.81 (0.67, 0.95) |
| Hard | 8300 | 5211,733 | 1.59 (1.56, 1.63) | 2.02 (1.69, 2.44) | 0.81 (0.66, 0.95) |
| Carpet | 717 | 6110,00 | 1.17 (1.09, 1.26) | 1.49 (1.23, 1.82) | 0.39 (0.22, 0.55) |
| **Round:** | | | | | |
| Final | 4307 | 2785,171 | 1.55 (1.50, 1.59) | 1 (Ref.) | 0 (Ref.) |
| Preliminary | 14600 | 9316,707 | 1.57 (1.54, 1.59) | 1.01 (0.98, 1.05) | 0.02 (-0.03, 0.07) |
| Qualifying | 407 | 2416,32 | 1.68 (1.52, 1.86) | 1.09 (0.98, 1.21) | 0.14 (-0.03, 0.31) |

**Table 9. WTA incidence rates, incidence rate ratio (IRR), risk difference, and the 95% confidence intervals (CI95%), expressed per 1000 games played.**

| WTA | Retirements | Games | Incidence Rate (95%CI) | IRR (95%CI) | RD (95%CI) |
|---|---|---|---|---|---|
| **Tournament Category:** | | | | | |
| WTA 125 Tournaments | 15 | 1083,8 | 1.38 (0.77, 2.28) | 1 (Ref.) | 0 (Ref.) |
| ITF Women's World Tennis Tour | 7291 | 5371,100 | 1.36 (1.33, 1.39) | 0.98 (0.59, 1.75) | -0.03 (-0.73, 0.67) |
| **Surface:** | | | | | |
| Grass | 124 | 1166,69 | 1.06 (0.88, 1.27) | 1 (Ref.) | 0 (Ref.) |
| Clay | 3509 | 2574,655 | 1.36 (1.32, 1.41) | 1.28 (1.07, 1.55) | 0.30 (0.11, 0.49) |
| Hard | 3380 | 2435,619 | 1.39 (1.34, 1.44) | 1.31 (1.09, 1.57) | 0.32 (0.13, 0.52) |
| Carpet | 293 | 2549,95 | 1.15 (1.02, 1.29) | 1.08 (0.87, 1.34) | 0.09 (-0.14, 0.31) |
| **Round:** | | | | | |
| Final | 1816 | 1260,368 | 1.44 (1.38, 1.51) | 1 (Ref.) | 0 (Ref.) |
| Preliminary | 5488 | 4120,616 | 1.33 (1.30, 1.37) | 0.92 (0.88, 0.98) | -0.11 (-0.18, -0.03) |
| Qualifying | 2 | 954 | 2.10 (0.25, 7.57) | 1.46 (0.18, 5.26) | 0.66 (-2.25, 3.56) |

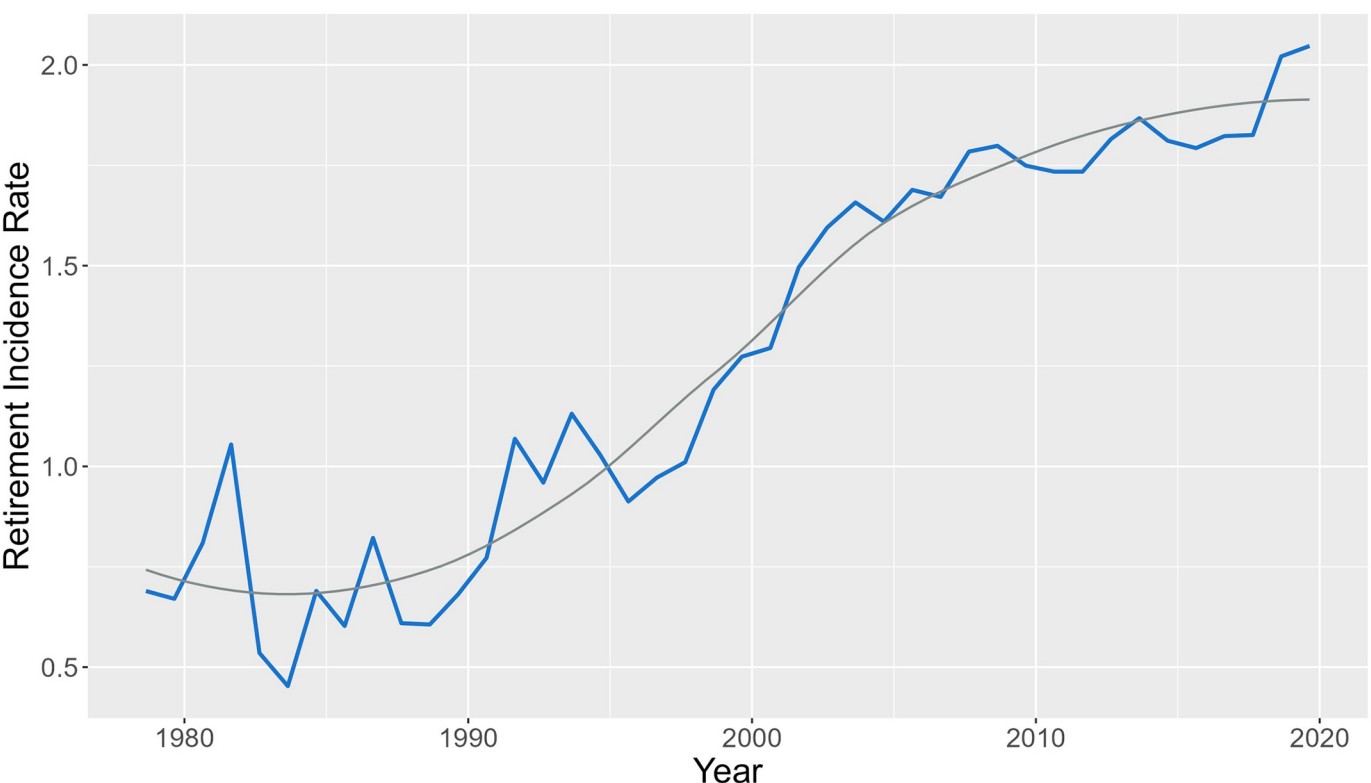

**Fig 1. Evolution of the incidence rate in the year-period 1978–2019 in ATP database, with a smooth line in grey.** The incidence rate is expressed as the number of retirements per 1000 games played.

epidemiological measures, together with the number of retirements and the number of games played, for the ATP database are shown in Table 8.

In ATP, the global incidence rate is 1.56 (95% CI from 1.54 to 1.59) retirements per 1000 games played. Focusing on the tournament category, the retirement incidence rate varies from 1.34 in ATP Challenger Tour to 1.65 in ITF Men's World Tennis Tour, being 1.23 times greater in the ITF Tour, with an absolute increase of 0.31 retirements per 1000 games played. Matches played on hard or clay courts have twice the risk of suffering a retirement (2.03 and 2.02, respectively) than when playing in grass courts, and therefore, the fact of playing in grass decreases the retirement incidence from 1.60 and 1.59 retirements per 1000 games played to 0.79. Specifically, the number of retirements is reduced in 0.81. The incidence rate in carpet surfaces is 1.17, being 1.49 times greater than in grass courts and increasing the retirement incidence by 0.39.

Depending on the match round, the incidence rate varies from 1.55 to 1.68. However, taking the final rounds as a reference, no relative association is found, as the 95% confidence intervals for the incidence risk ratio are (0.98, 1.05) and (0.98, 1.21) for preliminary and qualifying rounds, respectively. Likewise, no significant absolute association is found, being the 95% confidence intervals for the risk difference (-0.03, 0.07) and (-0.03, 0.31) for preliminary and qualifying rounds, respectively.

For the WTA database, the epidemiological measures, together with the number of retirements and the number of games played, are shown in Table 9. In WTA, the global incidence rate is 1.36 (95% CI from 1.33 to 1.39) retirements per 1000 games played. Focusing on the

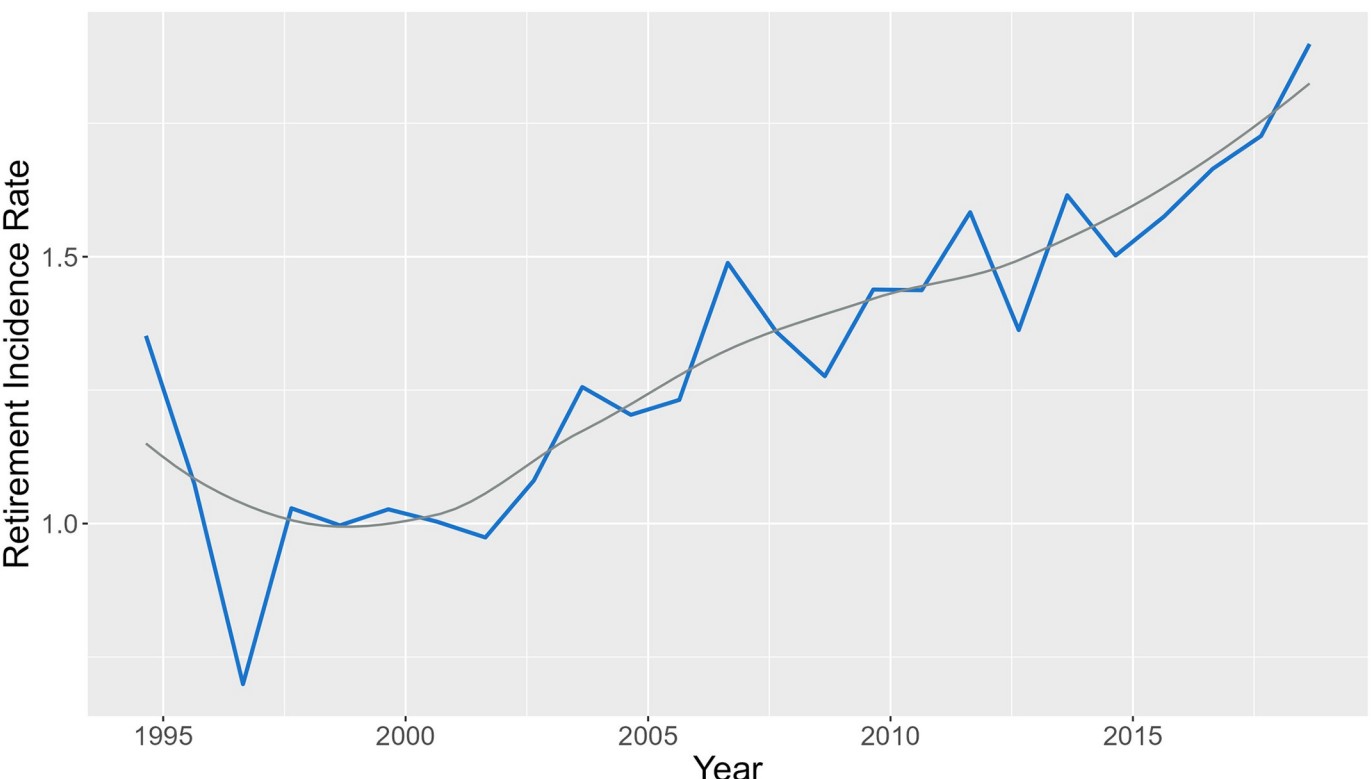

**Fig 2. Evolution of the incidence rate in the year-period 1994–2018 in WTA database, with a smooth line in grey.** The incidence rate is expressed as the number of retirements per 1000 games played.

tournament category, the incidence rate is 1.38 in WTA 125 Tournaments and 1.36 in the ITF Tour. However, the 95% confidence interval for the incidence rate ratio and risk difference is (0.59, 1.75) and (-0.73, 0.67), respectively, and therefore, no association is found.

Matches played on hard courts have 1.31 times more risk of suffering a retirement than matches played on grass courts with an incidence rate that varies from 1.39 to 1.06 retirements per 1000 games played, and reducing the number of retirements in 0.32. The risk of retiring in clay courts is 1.28 times greater than in grass surfaces, increasing the retirement incidence by 0.30 per 1000 games played. In carpets courts, the incidence rate is 1.15. However, no association is found, as the 95% confidence interval is (0.87, 1.34) for the incidence rate ratio and (-0.14, 0.31) for the risk difference.

Depending on the match round, the retirement incidence rate is 1.44 in final rounds and 1.33 in preliminary rounds, being 0.92 times less in preliminary rounds, and with an absolute reduction of -0.11. The incidence rate in qualifying rounds is 2.10, having 1.46 times more risk of retiring than in final rounds. However, no relative association is found as the 95% confidence interval is (0.18, 5.26). In the same way, no absolute association is found either as the 95% confidence interval is (-2.25, 3.56).

## Multivariable analysis

In this study, gender effect interaction with match characteristics covariates is analyzed (Fig 3).

Since the graphics suggest gender-round and gender-surface effect interactions and since male and female tennis competitions have their own idiosyncrasy, a different model for male

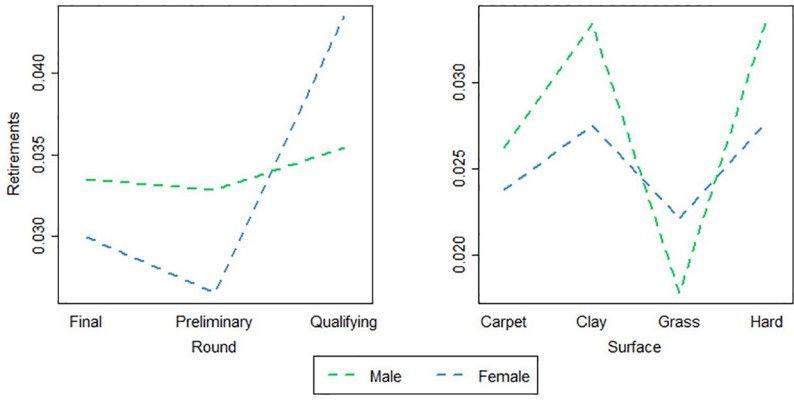

**Fig 3. Interaction plot between gender and match characteristics of round and surface of play.**

and female competitions is required, and therefore, a separately regression models were fitted for ATP and WTA.

**Poisson regression model in ATP.** Table 10 shows the Poisson regression model to assess risk factors of retirements in previous ATP Tour tournaments. According to the multivariable analysis, the variables significantly associated with retirements were tournament category, year, surface, and age mean.

**Poisson regression model in WTA.** The multivariable analysis using the Poisson regression model with offset to assess the risk factors of retirements in previous WTA Tour tournaments is shown in Table 11. The variables that remained significantly associated with risk of retirement in the final model were year, surface of play, and match round.

## Discussion

This study highlights the importance of epidemiological research in previous ATP and WTA Tour tournaments, which has traditionally been in the background. Our main findings show an increasing tendency of the retirement incidence rate over the years for both male and

**Table 10. Poisson regression model to analyze factors associated with retirements in previous ATP Tour tournaments.** * SE: Standard Error, ** IRR: Incidence Rate Ratio.

| Variables | Estimate | SE * | IRR (95% CI) ** | p-value |
|---|---|---|---|---|
| Intercept | -52.895 | 1.759 | | < 0.001 |
| Tournament Category | | | | |
| ATP Challenger | | | Ref. | Ref. |
| ITF Tour | 0.292 | 0.018 | 1.34 (1.29, 1.39) | < 0.001 |
| Year | 0.022 | 0.001 | 1.02 (1.02, 1.02) | < 0.001 |
| Surface | | | | |
| Grass | | | Ref. | Ref. |
| Carpet | 0.394 | 0.099 | 1.48 (1.23, 1.81) | < 0.001 |
| Clay | 0.619 | 0.092 | 1.86 (1.56, 2.24) | < 0.001 |
| Hard | 0.586 | 0.092 | 1.80 (1.51, 2.16) | < 0.001 |
| Mean Age (x5) | 0.271 | 0.013 | 1.31 (1.28, 1.35) | < 0.001 |

Mean_age(x5) represents the effect of a 5-year increase in the average age of the players.

**Table 11. Poisson regression model to analyze factors associated with retirements in WTA Tour tournaments.** * SE: Standard Error, ** IRR: Incidence Rate Ratio.

| Variables | Estimate | SE * | IRR (95% CI) ** | p-value |
|---|---|---|---|---|
| Intercept | -7.001 | 0.093 | | < 0.001 |
| I(Year^2) | -0.0006 | 0.0003 | 1.00 (1.00, 1.00) | 0.061 |
| Year | 0.039 | 0.006 | 1.04 (1.03, 1.05) | < 0.001 |
| Surface | | | | |
| Grass | | | Ref. | Ref. |
| Carpet | 0.073 | 0.107 | 1.08 (0.87, 1.33) | 0.497 |
| Clay | 0.208 | 0.091 | 1.23 (1.03, 1.48) | 0.023 |
| Hard | 0.212 | 0.092 | 1.24 (1.04, 1.49) | 0.021 |
| Round | | | | |
| Final | | | Ref. | Ref. |
| Preliminary | -0.078 | 0.027 | 0.93 (0.88, 0.98) | 0.004 |
| Qualifying | 0.186 | 0.708 | 1.20 (0.20, 3.72) | 0.793 |

female players in the sample of professional competitions studied. The variations in the occurrence of retirements have been found depending on the court surface with higher incidence in hard and clay courts than in grass for both genders. Additionally, the tournament category seemed to have an impact on the retirement incidence rate on male competitions while in female competitions the match round affected its incidence.

The increased tendency of retirements incidence rate over years found is consistent with other recent studies [38, 39] and suggests that the evolution of the game and the volume of play entail demanding physical requirements [9], influencing the increase of retirements. While these trends are reported in multiple studies, the comparison between incidence rates is difficult due to the range of different denominators used to express incidence rates in tennis epidemiology. Although this was not the objective of the study, as ATP and WTA competitions are independent and different, which makes it difficult to compare between genders, male competitions seem to have a higher incidence rate than female tournaments. These results are similar to those of other studies [20, 38, 74], in which the overall incidence was lower in female than in male players. Nevertheless, research findings are inconclusive as it has been reported higher injury rates for female players [30], as well as no evidence to suggest differences in retirement rates between men and women [20, 39]. All these discrepancies may be explained because of the differences between tournaments, and the distinct match characteristics and game styles between men and women, all of which affect the demands on players during match play [75, 76].

As already indicated, one important peculiarity of the sport is that is played on different surfaces, which may affect the players' game style and performance [14, 15], and require players to adjust to each surface sometimes within a short period of time, thus having an impact on injury rate [43]. Our results are consistent with these findings, showing variations in the occurrence of retirements depending on the court surface. Hard and clay courts showed a higher incidence rate compared to grass surfaces in both male and female tournaments. These findings are similar to those of other studies [77], and can be explained by the biomechanical differences, match characteristics, and friction properties of each surface [78]. Hard courts are believed to be the most high-risk surface because of the low shock absorption and high friction, producing a high impact force in athletes' joints, while clay courts are slow surfaces with lower ball speed, and therefore, the high retirement incidence may be explained as a result of longer rally duration and effective playing time. On the contrary, grass courts are fast surfaces, often

favoring shorter rally durations and more points concluded with high-speed shots, which may potentially lead to a higher incidence of abrupt movements and an increased risk of injuries, although it is worth noting that certain characteristics of grass courts, such as shorter points and a higher frequency of serves and net play, may mitigate the risk of injuries for players [9, 19, 79, 80].

As per the competition level, male tournaments seem to have an impact on the retirement incidence rate, with the ATP Challenger Tour having lower injury rates than the ITF Tour. Our findings are similar to those of Breznik and Batagelj (45,46)), who reported an inverse relationship between the importance of the tournament and the proportion of retirements, and with other recent studies (38). On the contrary, in female competitions, the tournament category did not affect the incidence rate of retirements. These results could be explained by the lack of data on WTA 125 Tournaments, although Néri-Fuchs et al. did not find any differences in the withdrawal rates between women tournament categories [38].

The tournament round where the match is played is also thought to be an influencing factor on the incidence of retirements, expecting lower rates in the final rounds of the tournaments. However, this was not found in our analyses.

Retirement incidence rates in men do not seem to be affected by the match round, as already concluded in other studies with professional male players [39, 46, 50]. In contrast, female competitions showed lower incidence rates in preliminary rounds when compared to final rounds. These results are consistent with those of Okholm Kryger et al., who found a retirement risk in women that increased up to the semifinals round [39].

With respect to the age of players, an age effect on the incidence of retirements has been reported in some studies focused on junior elite competitions. It's noteworthy that these studies encompass both male and female participants. The findings of these studies revealed that injury incidence increased with age [13, 52, 53]. In another study, Rice et al. compared injury rates between three age groups and found that adult and adolescent players had higher proportions of injuries than young athletes [44]. In our study, the effect of age on retirements was evaluated as the mean between the match players' ages. In the ATP, each 5-year increase in the average age of the match is associated with a 1.31-fold increase in the risk of retirements. However, no clear conclusions can be obtained from our analyses, as in female competitions age does not seem to have a significant effect, while in men seems to be related to the year the match was played.

The main novelty and contribution of our research is the fact that for the first time players' retirement is investigated on a sample of professional tournaments that had not been previously studied. However, this study has certain limitations that should be mentioned. The first of these is that the information available on certain variables in the professional tournaments studied is very scarce or null. It is also important to highlight that for this research a certain number of variables were identified and obviously it is possible to consider others that would provide a broader overview of the subject of study.

Future research on retirements in tournaments of previous stages to the ATP and WTA Tours is key due to the relevance of these events for future professional player development. New studies should benefit from consistent data collection, as the minutes played in a match are an important factor when reporting incidence rates. Furthermore, the analysis of new variables regarding match and player characteristics that can help draw a more precise picture of the risk factors behind a retirement. Finally, another possible line of research that would be very interesting could focus on the study of both male and female doubles because the information on the retirement of players in these modalities is practically non-existent.

## Conclusions

This study provides valuable insights into the increasing incidence of retirements in both male and female professional tennis over the observed period. Our analysis reveals a notable trend in the increasing incidence of retirements, particularly noticeable from around the end of the 90s, especially in the ATP tournament. Interestingly, while the evolution of retirement incidence rates follows an upward trajectory in both ATP and WTA tournaments, the incidence rates are consistently higher in ATP than in WTA. This finding suggests potential differences in the underlying factors contributing to retirements between male and female players or differences in the management of retirements across the two circuits. Our analysis also highlights the impact of playing surface and tournament category on retirement rates, with hard and clay courts exhibiting higher incidence rates compared to grass courts. Additionally, in men's tournaments, we observed a significant effect of tournament category on retirements, particularly lower rates in ATP Challenger competitions, while in women's tournaments, the match round emerged as a significant factor, with higher rates in final rounds. Overall, this study underscores the importance of understanding the underlying factors contributing to retirements in professional tennis and provides a foundation for developing targeted injury prevention strategies to optimize player performance.

## Author Contributions

**Conceptualization:** Maria Palau, Ernest Baiget, Jordi Cortés, Martí Casals.

**Data curation:** Maria Palau, Jordi Cortés, Martí Casals.

**Formal analysis:** Jordi Cortés, Martí Casals.

**Funding acquisition:** Martí Casals.

**Investigation:** Ernest Baiget, Martí Casals.

**Methodology:** Ernest Baiget, Jordi Cortés, Martí Casals.

**Resources:** Ernest Baiget, Martí Casals.

**Software:** Maria Palau, Jordi Cortés, Martí Casals.

**Supervision:** Maria Palau, Ernest Baiget, Jordi Cortés, Joan Martínez, Martí Casals.

**Validation:** Ernest Baiget, Joan Martínez, Miguel Crespo, Martí Casals.

**Visualization:** Jordi Cortés, Joan Martínez, Martí Casals.

**Writing – original draft:** Jordi Cortés, Martí Casals.

**Writing – review & editing:** Joan Martínez, Miguel Crespo, Martí Casals.

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
