## [Decision Letter · Decision Letter 0]

27 Mar 2024

PONE-D-24-04977Retirements of tennis players in professional tournaments of previous stages to the ATP and WTA ToursPLOS ONE

Dear Dr. Casals,

Thank you for submitting your manuscript to PLOS ONE. After careful consideration, we feel that it has merit but does not fully meet PLOS ONE’s publication criteria as it currently stands. Therefore, we invite you to submit a revised version of the manuscript that addresses the points raised during the review process.

We look forward to receiving your revised manuscript.

Kind regards,

Javier Abián-Vicén, Ph.D.

Academic Editor

PLOS ONE

Journal Requirements:

   "This research was funded by the Ministerio de Ciencia e Innovación (Spain) (PID2019- 352 104830RB-I00) and the Departament de Recerca i Universitats de la Generalitat de Catalunya (Spain) [2021 SGR 01421 (GRBIO)]."

Reviewers' comments:

Reviewer's Responses to Questions

**Comments to the Author**

1. Is the manuscript technically sound, and do the data support the conclusions?

Reviewer #1: Partly

Reviewer #2: Yes

2. Has the statistical analysis been performed appropriately and rigorously? 

Reviewer #1: I Don't Know

Reviewer #2: Yes

3. Have the authors made all data underlying the findings in their manuscript fully available?

Reviewer #1: Yes

Reviewer #2: Yes

4. Is the manuscript presented in an intelligible fashion and written in standard English?

Reviewer #1: Yes

Reviewer #2: Yes

5. Review Comments to the Author

Reviewer #1: The study aimed to use available epidemiological data to draw conclusions. While retirements of tennis players provided some useful insights, the results were not entirely conclusive. The statistical analysis was difficult to understand, hence more arguments were required to clarify how the conclusions were reached. It is crucial to explain the process of selecting a certain match level, the significance of each computed number, and how they led to the conclusions. I completely agree that this kind of study is necessary in the field of sports medicine to make the vast amount of available data meaningful and useful.

Reviewer #2: Retirements of tennis players in professional tournaments of previous stages to the ATP and WTA Tours

General comments

The aim of this study was to analyze epidemiological patterns and risk factors associated with the retirement of tennis players from previous professional tournaments. A retrospective cohort study was conducted. The study focused on past ATP (584,806 matches) and WTA (267,380 matches) tournaments in the period 1978-2019 for men and 1994-2018 for women. Potential risk factors such as playing surface, tournament category, round of play and player age were analyzed to assess withdrawals. The overall incidence rate was 1.56 withdrawals for men and 1.36 withdrawals for women per 1000 games played. The dropouts increased over the years. The incidence rates were different on hard court, clay court and grass. The risk factors differed according to gender and tournament round. This study provides coaches, players, support staff and epidemiologists with valuable insights into the breakdowns and associated risk factors at ATP and WTA tournaments, contributing to injury prevention strategies.

Special comments

Title

Line 1: The authors should consider whether the use of the term "of previous stages to the ATP and WTA Tours" really makes it clear to the reader which competitions they are referring to.

Introduction

Line 75: Otherwise, I think the concept of the introduction is well characterized, but I suggest that the authors add at least some parameters that better determine the workload of tennis players in different time periods (duration of matches, amount of exercise, number of rallies and strokes, calories burned, etc.). With this information, the causes of injuries can also be better explained. Perhaps they can also add the aspect of where injuries occur most frequently in tennis players, in tournaments or in training.

Line 138: Also in terms of conclusions, I suggest that the introduction includes a part that defines more precisely the evolution of workload in tennis for men and women during the observation period. In my opinion, the game of tennis has changed dramatically in 40 decades.

Materials and methods

Line 158: I suggest that in Table 4 the term “hand dominance” was used for both the winner and the loser of the match.

Line 291: The information “of which are completed matches and 7,306 (2.73%) are matches in which there was an abandonment” is reproduced in line 296.

Results

Line 314: In Table 5-9, a comma should be used for numbers above 9999.

Discussion

Line 487: The authors should explain what "preference for high-speed points" means.

Line 497: The authors should think about whether there is another reason why there is no difference in withdrawal rates in women's tournaments.

Line 509: is it possible that the age structure of tennis players who participated in tournaments during the observed period has also changed.

Conclusions

Line 532: Considering the observation period and the amount of data contained in the study, I think the conclusions are rather modest. As already mentioned, I suggest that the authors divide the observation period into smaller time intervals or find a more detailed explanation for the results obtained.

6. PLOS authors have the option to publish the peer review history of their article (what does this mean?). If published, this will include your full peer review and any attached files.

Reviewer #1: No

Reviewer #2: **Yes: **Ales Filipcic

---

## [Author Response · Author response to Decision Letter 0]

10 Apr 2024

Dear Editor,

We would like to thank you for giving us the opportunity of submitting a revision of the manuscript entitled “Retirements of players in professional tournaments of previous stages to the ATP and WTA Tours”.

We have carefully read the comments and suggestions that have certainly improved the manuscript. Additionally, a list of changes and answers to the reviewers’ comments has been submitted.

Note that the correct grant numbers for the awards of the study in the ‘Funding Information’ section are: "This research was funded by the Ministerio de Ciencia e Innovación (Spain) (PID2019- 352 104830RB-I00) and the Departament de Recerca i Universitats de la Generalitat de Catalunya (Spain) [2021 SGR 01421 (GRBIO)]. The funders had no role in study design, data collection and analysis, decision to publish, or preparation of the manuscript.

Sincerely,

Martí Casals

Sport and Physical Activity Studies Centre (CEEAF), University of Vic – Central University of Catalonia (UVic-UCC), C. Dr. Antoni Vilà Cañellas, s/n, 08500 Vic, Spain. 

ORCiD: https://orcid.org/0000-0002-1775-8331

Twitter: @CasalsTMarti

Response to Reviewers

We have carefully read the comments and suggestions that have certainly improved the manuscript. Additionally, a list of changes and answers to the reviewers’ comments has been submitted. 

Review Comments to the Author

Reviewer #1: The study aimed to use available epidemiological data to draw conclusions. While retirements of tennis players provided some useful insights, the results were not entirely conclusive. The statistical analysis was difficult to understand, hence more arguments were required to clarify how the conclusions were reached. It is crucial to explain the process of selecting a certain match level, the significance of each computed number, and how they led to the conclusions. I completely agree that this kind of study is necessary in the field of sports medicine to make the vast amount of available data meaningful and useful.

Thank you for the comment. We have clarified and updated the Conclusions subsection according to your suggestion (See below).

In addition, for a better statistical interpretation and seeing that the multivariable model of ATP (table 10) without the interaction of age, the conclusions were similar and equally parsimonious, it has been updated with the age multiplied by 5 to facilitate interpretation (which has been added to the Discussion section). 

“This study provides valuable insights into the increasing incidence of retirements in both male and female professional tennis over the observed period. Our analysis reveals a notable trend in the increasing incidence of retirements, particularly noticeable from around the end of the 90s, especially in the ATP tournament. Interestingly, while the evolution of retirement incidence rates follows an upward trajectory in both ATP and WTA tournaments, the incidence rates are consistently higher in ATP than in WTA. This finding suggests potential differences in the underlying factors contributing to retirements between male and female players or differences in the management of retirements across the two circuits. Our analysis also highlights the impact of playing surface and tournament category on retirement rates, with hard and clay courts exhibiting higher incidence rates compared to grass courts. Additionally, in men's tournaments, we observed a significant effect of tournament category on retirements, particularly lower rates in ATP Challenger competitions, while in women's tournaments, the match round emerged as a significant factor, with higher rates in final rounds. Overall, this study underscores the importance of understanding the underlying factors contributing to retirements in professional tennis and provides a foundation for developing targeted injury prevention strategies to optimize player performance.”

Reviewer #2: Retirements of tennis players in professional tournaments of previous stages to the ATP and WTA Tours

General comments

The aim of this study was to analyze epidemiological patterns and risk factors associated with the retirement of tennis players from previous professional tournaments. A retrospective cohort study was conducted. The study focused on past ATP (584,806 matches) and WTA (267,380 matches) tournaments in the period 1978-2019 for men and 1994-2018 for women. Potential risk factors such as playing surface, tournament category, round of play and player age were analyzed to assess withdrawals. The overall incidence rate was 1.56 withdrawals for men and 1.36 withdrawals for women per 1000 games played. The dropouts increased over the years. The incidence rates were different on hard court, clay court and grass. The risk factors differed according to gender and tournament round. This study provides coaches, players, support staff and epidemiologists with valuable insights into the breakdowns and associated risk factors at ATP and WTA tournaments, contributing to injury prevention strategies.

Special comments

Title

Line 1: The authors should consider whether the use of the term "of previous stages to the ATP and WTA Tours" really makes it clear to the reader which competitions they are referring to.

Thank you for your suggestion. We have updated the title: “Retirements of Professional Tennis Players in Second- and Third-Tier Tournaments on the ATP and WTA Tours”

Introduction

Line 75: Otherwise, I think the concept of the introduction is well characterized, but I suggest that the authors add at least some parameters that better determine the workload of tennis players in different time periods (duration of matches, amount of exercise, number of rallies and strokes, calories burned, etc.). With this information, the causes of injuries can also be better explained. Perhaps they can also add the aspect of where injuries occur most frequently in tennis players, in tournaments or in training.

Competition male and female tennis players have to face powerful strokes during matches up to 3 hours, with mean rallies durations and covered distances of 5.5 and 6.4 s and 9.6 m and 8.2 m [9]. Each rally requires to perform an average of 4-5 short, repetitive, and multidirectional high-intensity changes of direction, covering approximately 3 m, and resulting in a total of 250 to 400 changes of direction during the whole match [ref1]. Therefore, competition tennis players have to prepared to repeatedly over an extended period fo time execute hihg-intensity actions (i.e., shots and displacements) and to recover fast from it with both aerobic and anaerobic metabolic demands [ref2].

Ref1:

Giles, B., Peeling, P., & Reid, M. (2022). Quantifying change of direction movement demands in professional tennis matchplay: An analysis from the Australian Open Grand Slam. The Journal of Strength & Conditioning Research.

Ref 2:

Fernandez-Fernandez, J., Sanz-Rivas, D., & Mendez-Villanueva, A. (2009). A review of the activity profile and physiological demands of tennis match play. Strength & Conditioning Journal, 31(4), 15-26.

Line 138: Also in terms of conclusions, I suggest that the introduction includes a part that defines more precisely the evolution of workload in tennis for men and women during the observation period. In my opinion, the game of tennis has changed dramatically in 40 decades.

Thank you. We have added the following sentences to clarify:

"During the last decades, there has been a tendency towards increased ball speed and taller players, allowing professional players to generate higher amounts of power behind their shots [12]."

Materials and methods

Line 158: I suggest that in Table 4 the term “hand dominance” was used for both the winner and the loser of the match.

Done

Line 291: The information “of which are completed matches and 7,306 (2.73%) are matches in which there was an abandonment” is reproduced in line 296.

Thank you for your comment. We have deleted the paragraph from line 293 to 296, as it was redundant and repeated information as suggested.

Results

Line 314: In Table 5-9, a comma should be used for numbers above 9999.

Done

Discussion

Line 487: The authors should explain what "preference for high-speed points" means.

Thank you for your comment. We appreciate your concern and value your contribution to improving our article. After reviewing the literature and considering your observations, we acknowledge that the assertion regarding the relationship between high-speed points and the number of retirements on grass courts may be more nuanced than initially described. Therefore, we propose a revised wording of the sentence in question:

“On the contrary, grass courts are fast surfaces, often favoring shorter rally durations and more points concluded with high-speed shots, which may potentially lead to a higher incidence of abrupt movements and an increased risk of injuries, although it is worth noting that certain characteristics of grass courts, such as shorter points and a higher frequency of serves and net play, may mitigate the risk of injuries for players.”

Line 497: The authors should think about whether there is another reason why there is no difference in withdrawal rates in women's tournaments.

Thank you for the comment. While there are no differences based on tournament category, it's also unclear why there are different trends based on gender, and it would be worthwhile to address this in more detail in future studies.

Line 509: is it possible that the age structure of tennis players who participated in tournaments during the observed period has also changed.

Thank you for your comment. We have explored the age structure of tennis players participating in tournaments over the observed period. However, our analysis did not reveal any significant changes in age structure for both ATP and WTA players during the study period. Therefore, we did not address this aspect as a hypothesis in the Discussion section. We appreciate your suggestion and have included this clarification in the revised version of the manuscript.

Conclusions

Line 532: Considering the observation period and the amount of data contained in the study, I think the conclusions are rather modest. As already mentioned, I suggest that the authors divide the observation period into smaller time intervals or find a more detailed explanation for the results obtained.

Thanks. We have updated the Conclusions subsection according your suggestion. 

“This study provides valuable insights into the increasing incidence of retirements in both male and female professional tennis over the observed period. Our analysis reveals a notable trend in the increasing incidence of retirements, particularly noticeable from around the end of the 90s, especially in the ATP tournament. Interestingly, while the evolution of retirement incidence rates follows an upward trajectory in both ATP and WTA tournaments, the incidence rates are consistently higher in ATP than in WTA. This finding suggests potential differences in the underlying factors contributing to retirements between male and female players or differences in the management of retirements across the two circuits. Our analysis also highlights the impact of playing surface and tournament category on retirement rates, with hard and clay courts exhibiting higher incidence rates compared to grass courts. Additionally, in men's tournaments, we observed a significant effect of tournament category on retirements, particularly lower rates in ATP Challenger competitions, while in women's tournaments, the match round emerged as a significant factor, with higher rates in final rounds. Overall, this study underscores the importance of understanding the underlying factors contributing to retirements in professional tennis and provides a foundation for developing targeted injury prevention strategies to optimize player performance.”

---

## [Decision Letter · Decision Letter 1]

16 May 2024

Retirements of Professional Tennis Players in Second- and Third-Tier Tournaments on the ATP and WTA Tours

PONE-D-24-04977R1

Dear Dr. Casals,

We’re pleased to inform you that your manuscript has been judged scientifically suitable for publication and will be formally accepted for publication once it meets all outstanding technical requirements.

Kind regards,

Javier Abián-Vicén, Ph.D.

Academic Editor

PLOS ONE

Additional Editor Comments (optional):

Reviewers' comments:

Reviewer's Responses to Questions

**Comments to the Author**

1. If the authors have adequately addressed your comments raised in a previous round of review and you feel that this manuscript is now acceptable for publication, you may indicate that here to bypass the “Comments to the Author” section, enter your conflict of interest statement in the “Confidential to Editor” section, and submit your "Accept" recommendation.

Reviewer #1: All comments have been addressed

2. Is the manuscript technically sound, and do the data support the conclusions?

Reviewer #1: Yes

3. Has the statistical analysis been performed appropriately and rigorously? 

Reviewer #1: Yes

4. Have the authors made all data underlying the findings in their manuscript fully available?

Reviewer #1: Yes

5. Is the manuscript presented in an intelligible fashion and written in standard English?

Reviewer #1: Yes

6. Review Comments to the Author

Reviewer #1: Dear authors,

I would like to express my gratitude for submitting a revised article that addressed my questions and concerns. I also appreciated reading your explanations to the other reviewer, as it greatly improved my understanding of the topic. I sincerely hope that this article on the use of vast amounts of data in sports and competitions will provide valuable insights to clinicians, trainers, and other interested parties.

Thank you again for your hard work and dedication to this field.

Best regards, Nani Cahyani Sudarsono

7. PLOS authors have the option to publish the peer review history of their article (what does this mean?). If published, this will include your full peer review and any attached files.

Reviewer #1: **Yes: **Nani Cahyani Sudarsono

---

## [Editor Report · Acceptance letter]

21 May 2024

PONE-D-24-04977R1 

PLOS ONE

Dear Dr. Casals, 

I'm pleased to inform you that your manuscript has been deemed suitable for publication in PLOS ONE. Congratulations! Your manuscript is now being handed over to our production team.

Kind regards, 

on behalf of

Dr. Javier Abián-Vicén 

Academic Editor

PLOS ONE